# Abnormal Development of Microbiota May Be a Risk Factor for Febrile Urinary Tract Infection in Infancy

**DOI:** 10.3390/microorganisms11102574

**Published:** 2023-10-16

**Authors:** Chika Urakami, Sohsaku Yamanouchi, Takahisa Kimata, Shoji Tsuji, Shohei Akagawa, Jiro Kino, Yuko Akagawa, Shogo Kato, Atsushi Araki, Kazunari Kaneko

**Affiliations:** 1Department of Pediatrics, Kansai Medical University, Osaka 573-1010, Japan; urakamic@hirakata.kmu.ac.jp (C.U.); yamanous@hirakata.kmu.ac.jp (S.Y.); kimatat@hirakata.kmu.ac.jp (T.K.); tsujis@hirakata.kmu.ac.jp (S.T.); akagawas@hirakata.kmu.ac.jp (S.A.); akagaway@hirakata.kmu.ac.jp (Y.A.); katosh@hirakata.kmu.ac.jp (S.K.); 2Osaka Asahi Children’s Hospital, Osaka 535-0022, Japan; kinojr.co.jp@gmail.com (J.K.); araki@nakano-kodomo.or.jp (A.A.)

**Keywords:** febrile urinary tract infection, gut microbiota, abnormal development of microbiota, 16S rRNA gene sequencing

## Abstract

Febrile urinary tract infection (fUTI) is common in infants, but specific risk factors for developing it remain unclear. As most fUTIs are caused by ascending infections of intestinal bacteria, dysbiosis—an imbalance in gut microbial communities—may increase fUTI risk. This study was conducted to test the hypothesis that abnormal development of gut microbiota during infancy increases the risk of developing fUTI. Stool samples were collected from 28 infants aged 3–11 months with first-onset fUTI (fUTI group) and 51 healthy infants of the same age (HC group). After bacterial DNA extraction, 16S rRNA expression was measured and the diversity of gut microbiota and constituent bacteria were compared between the two groups. The alpha diversity of gut microbiota (median Shannon index and Chao index) was significantly lower in the fUTI group (3.0 and 42.5) than in the HC group (3.7 and 97.0; *p* < 0.001). The beta diversity also formed different clusters between the two groups (*p* < 0.001), suggesting differences in their microbial composition. The linear discriminant analysis effect size showed that the fUTI group proportionally featured significantly more *Escherichia-Shigella* in the gut microbiota (9.5%) than the HC group (3.1%; *p* < 0.001). In summary, abnormal gut microbiota development during infancy may increase the risk of fUTI.

## 1. Introduction

Urinary tract infection (UTI) is common in children, affecting 8.4% of girls and 1.7% of boys by the age of 7 years old; pediatric UTI accounts for 0.7% of physician visits annually. Thus, febrile UTI (fUTI) is one of the most common bacterial infections in children; among infants presenting with fever, the overall prevalence of UTI is 7.0% [1,2,3]. UTI is a burden for children and parents, and can cause short-term complications such as urosepsis [4].

fUTI is a diffuse pyogenic infection of the renal pelvis and parenchyma with symptoms including fever. However, diagnosing fUTI can be difficult because infants and young children with fUTI may only show nonspecific symptoms, such as poor appetite, failure to thrive, lethargy, irritability, vomiting, and diarrhea [4]. Up to 30% of infants have recurrent fUTI from 6 to 12 months after the first fUTI occurrence, and inappropriate management due to a delayed or missed diagnosis of fUTI or recurrent fUTI may lead to permanent renal scarring [4]. The incidence of renal scarring is reported to be 2.8–15.5% after one febrile UTI, 15.3–25.7% after two febrile UTIs, and 28.6–58.3% after three or more febrile UTIs [4,5,6,7]; it is thus clinically important to identify the risk factors for fUTI development. Although some risk factors for the development and recurrence of fUTI in infancy have been identified, such as congenital anomalies of the kidney and urinary tract (CAKUTs) including vesicoureteral reflux (VUR) and severe hydronephrosis [4,8], other risk factors remain unclear.

The advancements in genome sequencing technology have led to the human gut microbiota, which consists of more than 1000 species and 100 trillion gut bacteria, becoming widely recognized as an important organ [9,10,11]. As research on the gut microbiota has progressed, its balance has been shown to affect human health, with its imbalance being referred to as dysbiosis. Dysbiosis or abnormal development of the gut microbiota during infancy can have a variety of effects on the host’s health, most notably via aberrant immune responses, prolonged inflammation, increased production of toxic metabolites, and the dysregulation of metabolic and signaling pathways. This can in turn lead to the development of various chronic diseases, not only in adults, but also in children [9,10]. We have also reported that dysbiosis in gut microbiota is associated with pediatric diseases (e.g., idiopathic nephrotic syndrome, food allergies, and autism spectrum disorder in children born preterm) [11,12,13,14].

*Escherichia coli* is the most common bacterium responsible for fUTI in children, accounting for 84–92% of all cases [15,16,17]; indeed, most fUTIs are caused by ascending infections of enterobacteria such as *E. coli* [18]. Therefore, dysbiosis in the gut microbiota characterized by an enrichment of pathogenetic enterobacteria may be a risk factor for fUTI, but there are few reports supporting this. A previous study reported that *E. coli* in the intestinal microbiome was relatively more abundant in children with UTI than in controls [19]. However, patients with VUR, a well-known anatomically based risk factor for fUTI, were not excluded from that study.

Against this background, the present study was conducted to investigate whether the abnormal development of the gut microbiota increases the risk of developing fUTI in infants in the absence of anatomically based risk factors, such as CAKUTs, including VUR.

## 2. Materials and Methods

### 2.1. Study Participants

Thirty-nine infants aged 3–11 months diagnosed with first-episode fUTI were admitted to Osaka Asahi Children’s Hospital between April 2021 and March 2023. The criteria for diagnosing fUTI were high fever (≥38.5 °C) and detection of a single bacterium at ≥10^4^ cfu/mL in a urine sample collected through catheterization. Ultrasonography and voiding cystourethrography (VCUG) were performed in all patients to identify any CAKUTs, such as VUR and severe hydronephrosis. Ten patients with VUR and one with ureterocele were excluded from the study. There were no cases of severe hydronephrosis (3 degrees or higher in the Society for Fetal Urology classification). The 28 patients who did not have these CAKUTs were designated as the fUTI group. For a comparison group, 51 age- and sex-matched children who visited our hospital for immunization, who had no underlying disease, and whose parents gave consent for them to participate in the study were included [healthy control (HC) group]. The parents completed a questionnaire containing questions about the following factors that can affect gut microbiota: gestational age, mode of delivery, nutrition (breast, mixed, formula), probiotic intake, exposure to antibiotics, and the presence of siblings.

### 2.2. Sampling and Measurement

Stool samples for the HC group were collected when the subjects were well, and those for the fUTI group were collected on admission before the first antibiotic administration. After sampling, the samples were stored at −80 °C until DNA extraction. The samples were thawed within 1 week of stool collection and DNA was extracted using the NucleoSpin DNA Stool Kit (Macherey-Nagel, Düren, Germany). Sequencing of the 16S rRNA gene was performed by Macrogen Japan Inc. (Tokyo, Japan).

Sequencing libraries were prepared in accordance with Illumina’s 16S metagenomic sequencing library protocol to amplify the V3 and V4 regions. After extraction, 2 ng of stool sample gDNA was mixed with 5× reaction buffer, 1 mM dNTP mix, 500 nM Universal F/R PCR primers, and Herculase II fusion DNA polymerase (Agilent Technologies, Santa Clara, CA, USA), and PCR amplification was performed. Cycling conditions for the initial PCR were 3 min at 95 °C for thermal activation, 25 cycles of 30 s at 95 °C, 30 s at 55 °C, and 30 s at 72 °C, followed by final extension for 5 min at 72 °C. The universal primer pair with Illumina adapter overhang sequence used for the first amplification was as follows: V3-F, 5′-tcgtcgcgcagcgtcagatggtagtataagacagcctacgggnggcwgcag-3′; and V4-R, 5′-gtctctcgtgcgctctcgagatgatagatagagacaggactachvgggtactaatcc-3′. The PCR products from the first round were purified with AMPure beads (Agencourt Bioscience, Beverly, MA, USA). After purification, 2 µL of the first-round PCR product was amplified in a second round of PCR, and a final library, including the index, was constructed using the NexteraXT Indexed Primer. The cycling conditions for the second round of PCR were the same as in the first round, except that the number of cycles in the amplification step was reduced to 10. Again, the PCR products were purified with AMPure beads. The final purified product was quantified through qPCR in accordance with the qPCR Quantification Protocol Guide (KAPA Library Quantification kits for Illumina Sequencing platforms) and qualified using apeStation D1000 ScreenTape (Agilent Technologies, Waldbronn, Germany).

Sequence reads were imported into the Quantitative Insights Into Microbial Ecology version 2 (QIIME2) pipeline (version 2021.12) for bacterial identification and diversity analysis [20]. DADA2 [21] was used for quality filtering and feature [operational taxonomic unit (OTU)] prediction. After checking the sequence quality data, 30 nucleotides (nt) were trimmed from the 5′ end of forward reads and 30 nt from reverse reads; forward reads were trimmed to 280 nt and reverse reads to 240 nt. OTUs/features were classified using a pre-trained naïve Bayes classifier. Classifiers were trained using “Silva 138 99% OTUs” [22].

A table of the number of classifications and percentages (relative frequencies) was generated. Diversity analysis was performed on the resulting OTU/feature biom tables to provide systematic and non-systematic indices of alpha and beta diversity [23]. The visualization file (.qzv) can be viewed at “http://view.qiime2.org (accessed on 27 September 2023)”. The sequences were then clustered into OTUs with a threshold of 97% identity. Rarefaction was performed on all sample sequences to a maximum depth of 10,000 sequences. Alpha diversity and beta diversity were measured to evaluate the differences in gut microbiota and dysbiosis in gut microbiota between the two groups. Alpha diversity was calculated using the Shannon index and beta diversity was calculated by the Bray Curtis dissimilarity using the PERMANOVA test. The matrix of Bray Curtis dissimilarity was transformed into a new set of orthogonal axes, with the maximum variation factor indicated by the first principal coordinate axis and the second maximum variation factor by the second principal coordinate axis. The effect size (LEfSe) of linear discriminant analysis (LDA) was used to analyze differences in abundance. LEfSe combines a standard test for statistical significance with an additional test that encodes the association between biological consistency and effect to determine the feature most likely to explain differences between classes (organism, clade, OTUs, gene, or function). The LDA score was set at 4.0.

### 2.3. Statistical Analysis

Group comparisons were performed using the chi-squared test and Fisher’s exact test for qualitative data, and the Mann–Whitney U test for numerical data. All median and quartile values were calculated, and *p* < 0.05 was considered statistically significant. G*Power version 3.1.9.4 (Heinrich Heine University, Düsseldorf, Germany) was used to calculate the appropriate sample size for this study [24]. The appropriate sample sizes for a type I error of 0.05, power of 0.8, allocation ratio of 2, and effect size of 0.65 were 24 in the fUTI group and 48 in the HC group. Other statistical analyses were performed using BellCurve for Excel (version 3.21; Social Research and Information, Inc., Tokyo, Japan).

### 2.4. Institutional Review Board Statement

This study was conducted in accordance with the tenets of the Declaration of Helsinki and approved by the Ethics Committee of Osaka Asahi Children’s Hospital (No. 52-2). Informed consent was obtained from the guardians of all participants prior to enrollment.

## 3. Results

### 3.1. Profiles of Study Participants

A summary of the fUTI and HC groups is shown in Table 1. The fUTI group and HC group consisted of 28 patients (14 boys and 14 girls, median age 5 months, interquartile range (IQR) 3.8–7.0 months) and 51 children (28 boys and 23 girls, median age 5 months, IQR 4.0–8.0 months), respectively. According to the results of the urinary culture, *E. coli* was detected in 27 cases and *Citrobacter* in one case. There were no significant differences in the age in months, sex, gestational age, mode of delivery, nutrition, exposure to antibiotics within 1 month prior to infection, probiotic intake, and the presence of siblings between the two groups. With regard to the perinatal factors that can affect the gut microbiota, gestational age and mode of delivery were investigated. Gestational age (median, IQR) was not significantly different between the fUTI group (39.2, 38.1–40.1) and the HC group (39.6, 38.4–40.4). Mode of delivery (vaginal:cesarean) was also not significantly different between the fUTI group (23:5) and the HC group (43:8). There was also no significant difference in the type of nutrition (breast:mixed:formula) between the fUTI group (8:20:0) and the HC group (7:44:0), although the percentage of breastfed infants in the fUTI group (8 of 28: 28.5%) was greater than that in the HC group (7 of 51: 13.7%). There was no significant difference in the presence of siblings between the fUTI group (*n* = 15/28; 53.6%) and the HC group (*n* = 19/51; 37.3%). With regard to the factors that directly alter the gut microbiota, the use of antibiotics within 1 month before sample collection and the administration of probiotics or yogurt within 1 month before sample collection (at least 2 days a week) were evaluated. None of the infants in the fUTI group or the HC group used antibiotics within 1 month before sample collection. The percentage of infants with the administration of probiotics or yogurt within 1 month before sample collection (at least 2 days a week) was not significantly different between the fUTI group (*n* = 3/28; 10.7%) and the HC group (*n* = 7/51; 13.7%). With regard to allergic diseases, the presence of asthma, atopic dermatitis, and allergic rhinitis was surveyed. There was no significant difference in the rate of these conditions between the fUTI group (*n* = 2/28; 7.1%) and the HC group (*n* = 1/51; 2.0%).

### 3.2. Alpha Diversity

The Shannon and Chao indices of the two groups are shown in Figure 1A,B, respectively. Both indices were significantly lower in the fUTI group than in the HC group [Shannon index: median 3.0 (IQR: 2.7–3.5) in the fUTI group, median 3.7 (IQR: 3.2–4.6) in the HC group, *p* < 0.001; Chao index: median 42.5 (IQR: 33.5–48.5) in the fUTI group, median 97.0 (IQR: 69.5–132.0) in the HC group, *p* < 0.001].

### 3.3. Beta Diversity

To assess the differences in the gut microbiota between the fUTI and HC groups, principal coordinate analysis (using the Bray–Curtis dissimilarity to assess beta diversity) was used to characterize the samples in two dimensions. Each group formed a distinct cluster (Figure 2, *p* < 0.001), suggesting a different gut microbiota composition in each group.

### 3.4. LEfSe Analysis

The LEfSe (LDA score size = 4.0) analysis showed that the fUTI group had a greater abundance of the genus *Escherichia-Shigella* belonging to the order Enterobacterales and the family Enterobacteriaceae than the HC group (Figure 3). By contrast, the HC group had greater abundances of *Bacteroides fragilis* belonging to the phylum Bacteroidota, the class Bacteroidia, the order Bacteroidales, the family Bacteroidaceae, and the genus *Bacteroides* than the fUTI group. Table 2 shows the representative bacterial taxa with significant differences in abundance between the groups at the phylum, class, order, family, and genus levels.

### 3.5. Relative Abundance of Genus Escherichia-Shigella in Gut Microbiota

The proportion of the genus *Escherichia-Shigella*—which includes *E. coli*, the main causative bacterium of fUTI—in the gut microbiota was significantly higher in the fUTI group (median 9.5%, IQR 4.6–22.2%) than in the HC group (median 3.1%, IQR 1.2–6.5%; Figure 4, *p* < 0.001).

## 4. Discussion

To the best of our knowledge, this is the first report to show that a high percentage of *Escherichia-Shigella* in the gut microbiota of infants may be a risk factor for developing fUTI in the absence of CAKUTs. This is based on the following findings: When the gut microbiota of the fUTI and HC groups was compared, significant differences were found in both the alpha and beta diversity. In addition, the proportion of *Escherichia-Shigella* in the gut microbiota was significantly higher in the fUTI group than in the HC group. In general, *E. coli* comprises the majority of *Escherichia-Shigella*; the results thus suggest that the fUTI group had a higher proportion of *E. coli* in their gut microbiota than the HC group.

To date, three reports stating that dysbiosis in gut microbiota is associated with the development of a UTI have been published [19,25,26]. For example, in adults who had undergone a kidney transplant, dysbiosis in gut microbiota increased the risk of developing a UTI [25,26], especially when the relative abundance of Enterococcus was ≥1% [26]. Only one reported study compared the gut microbiota in children with a first fUTI and healthy controls [19]. In contrast to our results, that study did not find any significant difference in the alpha diversity, an indicator of dysbiosis in gut microbiota, while the beta diversity was not assessed. In addition, the differences in the relative abundance and mean quantity of *E. coli* between the UTI patients and controls were not statistically significant, although they tended to be higher in the patient group [19]. However, that study did not evaluate VUR, the most common risk factor for the onset and recurrence of fUTI complications. By contrast, we performed VCUG on all of the patients in the fUTI group to exclude patients with VUR, the most common anatomical risk factor for fUTI. Therefore, the differences between our results and the previous study could have resulted from the confounding influence of VUR.

For alpha diversity, both the Shannon and Chao indices were significantly lower in the fUTI group than in the HC group. The Shannon index represents the evenness of microbiota, while the Chao index indicates its richness. In low-diversity microbiota, the relative abundance of beneficial bacteria that ferment complex sugars to short-chain fatty acids, including butyrate, markedly decreases [27]. Butyrate has been shown to have both local and systemic anti-inflammatory effects; thus, its loss may mediate immune phenotypes in disease. Thus, it might be considered that fUTI is partly attributable to low-diversity microbiota.

The results of our LEfSe analysis showed that the fUTI group had a significantly lower proportion of *Bacteroides fragilis* in the gut microbiota than the HC group. Animal studies have shown that *B. fragilis* protects against ulcerative colitis, bronchial asthma, inflammation of the lungs, autoimmune encephalitis, and colorectal cancer [28,29]. Mechanistically, polysaccharide A from *B. fragilis* induces the inhibition of NF-κB, induction of regulatory T cells, and suppression of proinflammatory helper T cells [28,29]. As a result, inflammatory cytokines are suppressed and the clinical condition improves. Thus, the current study indicates that the association between the abnormal development of gut microbiota during infancy and fUTI results from both an increase in the relative abundance of *E. coli*, which can physically ascend the urinary tract, and a decrease in the relative abundance of *B. fragilis*, which impairs the systemic immune defense function. Meanwhile, the LEfSe analysis unexpectedly showed that the fUTI group had a greater abundance of the genus *Bifidobacterium*, which has been demonstrated to benefit the host by accelerating the maturation of the immune response, balancing the immune system to suppress inflammation, improving the intestinal barrier function, and increasing acetate production [30]. Considering that the genus *Bifidobacterium* has been demonstrated to predominate in the gut microbiota of breastfed infants, its higher abundance in the fUTI group might be explained by the type of nutrition: the percentage of breastfed infants in the fUTI group (8 of 28: 28.5%) was greater than that in the HC group (7 of 51: 13.7%), as shown in Table 1. It is assumed that this higher abundance of *Bifidobacterium* alone is insufficient to suppress the growth of *Escherichia-Shigella*, which are the causative bacteria of fUTI.

This study has several limitations. First, because this study was cross-sectional in nature, causal relationships were difficult to prove. Although stool was collected before the administration of antibiotics, there were concurrent changes in the gut microbiota composition and UTI. In this regard, recent observations in cases of fecal microbiota transplantation (FMT) may provide an insight into the causal relationship between gut dysbiosis and fUTI development: FMT is thought to improve the clinical state of each disease by correcting dysbiosis in gut microbiota [31,32], suggesting that dysbiosis in gut microbiota is a pathogenic factor rather than a consequence of each disease. In fact, FMT has been used to treat various diseases, such as recurrent *Clostridioides difficile* infection (rCDI), ulcerative colitis, multiple sclerosis, autism spectrum disorder, systemic lupus erythematosus, acute myeloid leukemia, graft-versus-host disease, and diabetes mellitus [31,32]. In addition, FMT in adult female patients with recurrent UTI (rUTI) prevents the recurrence of UTI [33,34,35,36]. Again, the mechanism preventing UTI recurrence may be an improvement in dysbiosis of gut microbiota following FMT [33,34,35,36]. Specifically, increased alpha diversity and a decreased relative abundance of uropathogens in the patient’s gut microbiota have been reported after FMT. Aira et al. reported that FMT cured rCDI and rUTI in a 93-year-old woman with rUTI complicated by rCDI [34]. At baseline, in this patient, the uropathogen Enterobacteriaceae including the genus *Escherichia* accounted for 74% of the gut microbiota, but after FMT, Enterobacteriaceae was markedly reduced to 0.07%; in addition, the Shannon index was 4.60 before FMT, increasing to 6.42 after FMT, indicating an improvement in the diversity. These findings were maintained for almost 1 year, with no recurrence of CDI or UTI. Based on these reports, the abnormal development of gut microbiota during infancy revealed in the present study is likely a risk factor for, rather than a consequence of, fUTI development. Second, we could not identify the underlying cause of the abnormal development of gut microbiota during infancy; some well-known risk factors for dysbiosis during the fetal period and infancy include premature birth, cesarean delivery, formula feeding, exposure to antibiotics, and the presence of siblings [11]. However, in the current study, no significant differences were found between the fUTI and HC groups for these risk factors. Other risk factors include weaning food intake, genetic background, and dysbiosis in the mother’s gut and vaginal microbiota [11]. We did not investigate these risk factors; therefore, we cannot draw conclusions about whether they caused dysbiosis in the fUTI group. Third, sequencing of the 16S rRNA gene was performed only at the V3–V4 regions in the current study. A previous study comparing 16S full-length synthetic long-read sequencing data with short-read (V3–V4) data disclosed that they were highly similar at all classification resolutions except the species level. At the species level, the 16S full-length synthetic long-read data showed better resolution than the short-read data in the analyses [37]. Thus, in this study, the analysis of the gut microbiota was performed up to the genus level, and there was no direct evidence that the fUTI group had a higher proportion of *E. coli* in the gut microbiota. However, taking into consideration the fact that most bacteria belonging to *Escherichia-Shigella* are *E. coli*, it is likely that the fUTI group had a higher proportion of *E. coli* than the HC group. Nonetheless, it is worth performing analysis through 16S full-length synthetic long-read sequencing at the species level.

## 5. Conclusions

We found clear differences in the gut microbiota between infants with fUTI and healthy controls. The infants with fUTI exhibited an abnormal development of gut microbiota during infancy, characterized by low diversity, a high proportion of *Escherichia-Shigella*, and a low proportion of *B. fragilis*. Therefore, this may be a risk factor for the development of fUTI, particularly in children without CAKUTs. Nonetheless, further research is clearly needed to confirm the causality before gut microbiota can be used as a therapeutic target to prevent recurrent fUTIs.

If the causal relationship is proven, interventions to correct the abnormal development of gut microbiota during infancy, such as probiotics, prebiotics, and synbiotics, may reduce the risk of fUTI in infants. In fact, many researchers, including those in our laboratory, have reported an improvement in the clinical states of various diseases using these methods to correct dysbiosis in the gut microbiota [11,38].

## Figures and Tables

**Figure 1 microorganisms-11-02574-f001:**
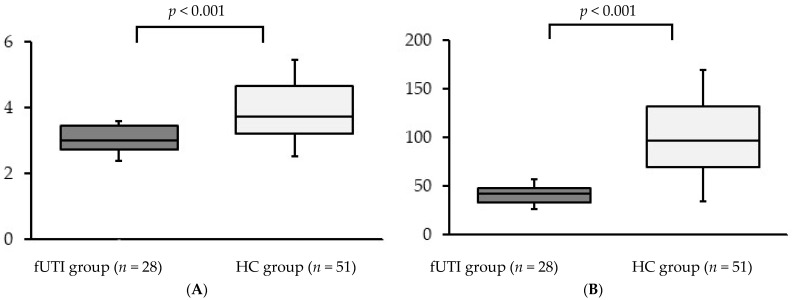
Alpha diversity of the gut microbiota of the fUTI and HC groups. (**A**) Shannon index; (**B**) Chao index. The central horizontal line represents the median, while the bottom and top edges of the box indicate the 25th and 75th percentiles, respectively. The vertical lines in the center represent the 5th and 95th percentiles. fUTI: febrile urinary tract infection, HC: healthy control.

**Figure 2 microorganisms-11-02574-f002:**
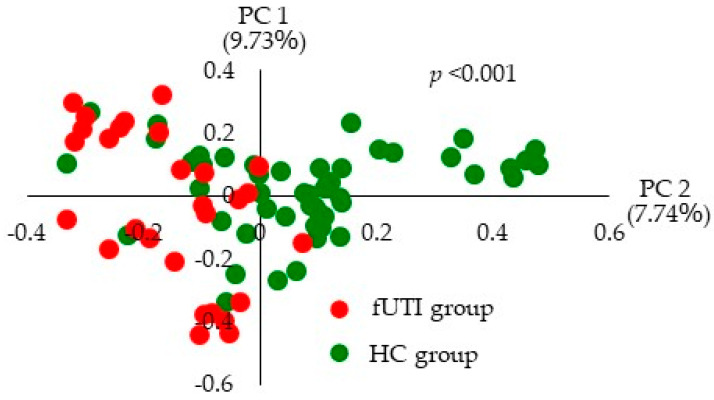
Principal coordinate analysis plot of the Bray–Curtis dissimilarity in the fUTI and HC groups. Red dots indicate clusters of the fUTI group and green dots indicate clusters of the HC group.

**Figure 3 microorganisms-11-02574-f003:**
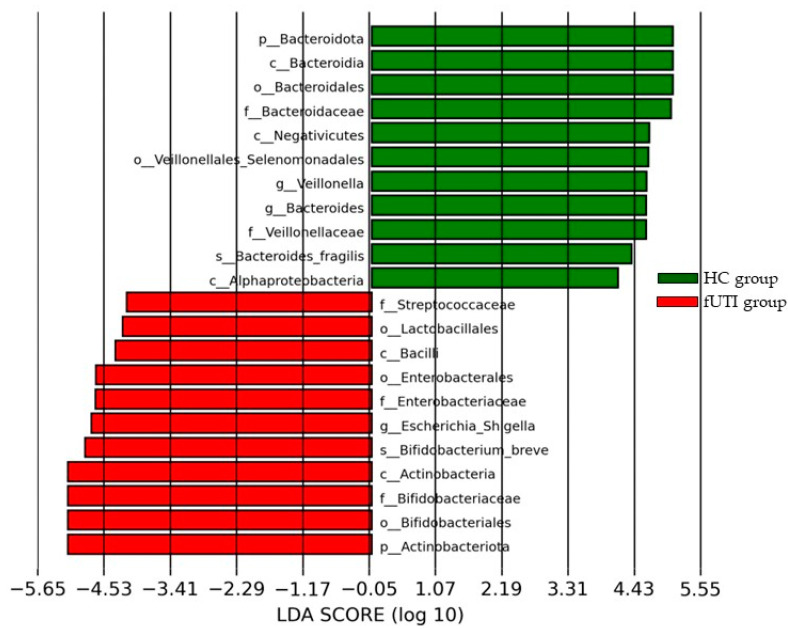
Differences in the dominant bacteria between the fUTI and HC groups according to LEfSe. LDA scores were calculated for taxa that showed significant differences in the type and abundance of microorganisms between the fUTI (red) and HC (green) groups. LDA scores are shown on a log10 scale.

**Figure 4 microorganisms-11-02574-f004:**
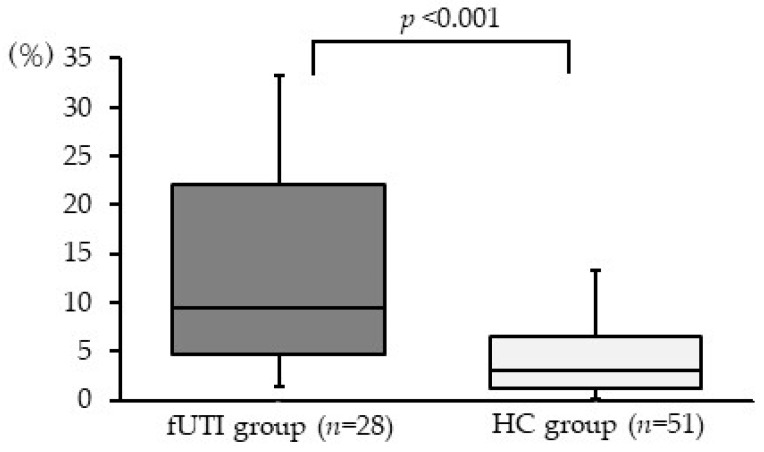
Relative abundance of the genus *Escherichia-Shigella* in the gut microbiota of fUTI and HC groups. This genus accounted for a significantly higher proportion of the gut microbiota in the fUTI group than in the HC group (*p* < 0.001). The central horizontal line represents the median, while the bottom and top edges of the box indicate the 25th and 75th percentiles, respectively. The vertical lines in the center represent the 5th and 95th percentiles, respectively.

**Table 1 microorganisms-11-02574-t001:** Participants’ Profiles.

	fUTI Group (*n* = 28)	HC Group (*n* = 51)	*p* Value
Age, months (median, IQR)	5 (3.8–7.0)	5 (4.0–8.0)	0.40
Sex (Boy: Girl)	14:14	28:23	0.86
Gestational age, weeks (median, IQR)	39.2 (38.1–40.1)	39.6 (38.4–40.4)	0.20
Mode of delivery (vaginal: cesarean)	23:5	43:8	1.00
Nutrition (breast: mixed: formula)	8:20:0	7:44:0	0.14
Siblings	15 (53.6)	19 (37.3)	0.24
Use of antibiotics within 1 month before sample collection (%)	0 (0)	0 (0)	1.00
Administration of probiotics or yogurt within 1 month before sample collection (%) (at least two days a week)	3 (10.7)	7 (13.7)	1.00
Allergy-related diseases (%) ^†^	2 (7.1)	1 (2.0)	0.29

fUTI group: febrile urinary tract infection group, HC group: healthy control group, IQR: interquartile range. ^†^: Allergy-related diseases include asthma, atopic dermatitis, allergic rhinitis, and food allergies.

**Table 2 microorganisms-11-02574-t002:** Representative bacterial taxa with significant differences in abundance at various taxonomic levels.

	fUTI Group (*n* = 28)	HC Group (*n* = 51)
Phylum	Actinobacceriota	Bacteroidota
Class	Actinobacteria	Bacteroidia
	Bacilli	Negativicutes
Order	Bifidobacteriales	Bacteroidales
	Enterobacterales	Veillonellases Selenomonadales
Family	Bifidobacteriaceae	Bacteroidaceae
	Enterobacteriaceae	Veillonellaceae
Genus	*Escherichia Shigella*	*Veillonella*
		*Bacteroides*

## Data Availability

The datasets used and analyzed in this study are located in Kansai Medical University Research Data Storage and are available from the corresponding author upon reasonable request. Owing to privacy restrictions, the data are not publicly available.

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
