# Peer review of "Abnormal Development of Microbiota May Be a Risk Factor for Febrile Urinary Tract Infection in Infancy"

_microorganisms, 2023, doi:10.3390/microorganisms11102574_

Round 1

Reviewer 1 Report

The authors investigated whether dysbiosis in gut microbiota increases the risk of developing fUTI in infants. This study recruited 28 infants aged 3–11 months with first-onset fUTI and 51 healthy infants. Results from this study demonstrated that alpha diversity was lower in the fUTI group than in the HC group. The beta diversity showed different clusters between the two groups. They found fUTI group had a higher abundance of genus Escherichia-Shigella than that in the HC group. The authors concluded that dysbiosis of gut microbiota during infancy can increase the risk of developing fUTI. I have several issues I feel need to be addressed and would welcome the author’s comments on these.

Major issues:

1. There are several biological taxonomy including species, genus, family, order, class, phylum, kingdom, and domain. The authors had better discuss whether other levels of gut microbiota are the same or different between two groups. For example, the ratio of Firmicutes and Bacteroidetes, the major phyla of the colon, has been related to several diseases. Did the fUTI group have a higher F/B ratio?

2. In the current study, sequencing of the 16S rRNA gene was performed in V3-V4 regions. However, comparison of the 16S rRNA full-length data with data derived 16S rRNA V4 region revealed differences in relative bacterial abundances, and α- and β-diversity. The use of full-length analysis should be taken into consideration for future study.

3. LefSe analysis showed the fUTI group had a greater abundance of Bifidobacterium. As Bifidobacterium is a well-known probiotic, why its abundance is high but not low in the disease group?

4. Why CAKUT group was excluded? It is inserting to see whether gut microbiota composition is different between three groups (HC, fUTI, and CAKUT).

5. The authors conclude that “Dysbiosis of gut microbiota during infancy can increase the risk of developing fUTI.” As this is a cross-sectional study, cause-and-effect relationship is hard to be proven. Although stool was collected before antibiotics administration, the changes in gut microbiota composition and UTI exist concurrently. In other words, fUTI might be a risk for gut microbiota dysbiosis. Please be careful to not overstate your point or make absolute conclusion that are not supported by evidence.

Minor issues:

1. Table 1. Please add weeks to gestational age.

2. Figure 1. Usually, the p value should be expressed to 3 digits. Please change p value to <0.001.

3. Figure 2. Usually, the p value should be expressed to 3 digits. Please change p value to <0.001.

I hope that the enclosed comments will be of help to the authors. 

Author Response

Reviewer 1

Comments and Suggestions for Authors

The authors investigated whether dysbiosis in gut microbiota increases the risk of developing fUTI in infants. This study recruited 28 infants aged 3–11 months with first-onset fUTI and 51 healthy infants. Results from this study demonstrated that alpha diversity was lower in the fUTI group than in the HC group. The beta diversity showed different clusters between the two groups. They found fUTI group had a higher abundance of genus Escherichia-Shigella than that in the HC group. The authors concluded that dysbiosis of gut microbiota during infancy can increase the risk of developing fUTI.

I have several issues I feel need to be addressed and would welcome the author’s comments on these.

Response

We thank the reviewer for this careful assessment of our manuscript and the useful comments provided. The comments of the reviewers were highly insightful and enabled us to greatly improve the quality of our manuscript. We have carefully revised our paper in response to these suggestions. In the following pages are our point-by-point responses to each of the reviewers’ comments. Modified sections are indicated by bold font and underlining in the revised manuscript.

Major issues:

  1. There are several biological taxonomy including species, genus, family, order, class, phylum, kingdom, and domain. The authors had better discuss whether other levels of gut microbiota are the same or different between two groups. For example, the ratio of Firmicutes and Bacteroidetes, the major phyla of the colon, has been related to several diseases. Did the fUTI group have a higher F/B ratio?

Response

Thank you for this suggestion. Representative bacterial taxa with significant differences in abundance between the fUTI group and the HC group at the levels of phylum, class, order, family, and genus are listed in Table 2 in the revised manuscript.

Regarding the ratio of Firmicutes to Bacteroides, which is known as a biomarker for obesity in children (Indiani CMDSP, et al. Childhood Obesity and Firmicutes/Bacteroidetes Ratio in the Gut Microbiota: A Systematic Review. Child Obes. 2018 Nov/Dec;14(8):501-509. doi: 10.1089/chi.2018.0040. PMID: 30183336), it was 0.19 for the fUTI group and 1.23 for the HC group. However, this index was not described in the text as it is unknown whether this ratio is associated with urinary tract infection. We appreciate your understanding on this issue.

  1. In the current study, sequencing of the 16S rRNA gene was performed in V3-V4 regions. However, comparison of the 16S rRNA full-length data with data derived 16S rRNA V4 region revealed differences in relative bacterial abundances, and α- and β-diversity. The use of full-length analysis should be taken into consideration for future study.

Response

We completely agree with your comment that the use of full-length analysis should be considered for future study. The previous study comparing 16S full-length synthetic long-read sequencing data with short-read (V3–V4) data disclosed that they were highly similar at all classification resolutions, except the species level. At the species level, 16S full-length synthetic long-read data showed better resolution than short-read data in analyses (Jeong J, et al. The effect of taxonomic classification by full-length 16S rRNA sequencing with a synthetic long-read technology. Sci Rep. 2021 Jan 18;11(1):1727. PMID: 33462291). These points have been added to the limitations with the related citation in the revised version of the manuscript.

  1. LefSe analysis showed the fUTI group had a greater abundance of Bifidobacterium. As Bifidobacterium is a well-known probiotic, why its abundance is high but not low in the disease group?

Response

As you pointed out, LEfSe analysis unexpectedly showed that the fUTI group had a higher abundance of the genus Bifidobacterium, which has been demonstrated to benefit the host by accelerating maturation of the immune response, balancing the immune system to suppress inflammation, improving intestinal barrier function, and increasing acetate production (Chichlowski M, et al. Bifidobacterium longum Subspecies infantis (B. infantis) in Pediatric Nutrition: Current State of Knowledge. Nutrients. 2020 May 28;12(6):1581. doi: 10.3390/nu12061581. PMID: 32481558). Considering that the genus Bifidobacterium has been demonstrated to predominate in the gut microbiota of breastfed infants, the higher abundance of this genus in the fUTI group might be explained by the type of nutrition: the percentage of breastfed infants in the fUTI group (8 of 28: 28.5%) was greater than that in the HC group (7 of 51: 13.7%), as shown in Table 1. It is assumed that this higher abundance of Bifidobacterium alone is insufficient to suppress the growth of Escherichia-Shigella, which are the causative bacteria of fUTI. This information was added in the discussion section.

  1. Why CAKUT group was excluded? It is inserting to see whether gut microbiota composition is different between three groups (HC, fUTI, and CAKUT).
  2. Response

Thank you for your suggestion. A previous study has already reported that E. coli in the intestinal microbiome was relatively more abundant in children with UTI than in controls, although the difference was insignificant [Paalanne, N, et al. Intestinal microbiome as a risk factor for urinary tract infections in children. Eur. J. Clin. Microbiol. Infect. Dis. 2018, 37, 1881-1891]. The authors of this study did not exclude patients with VUR, which is a well-known anatomically based risk factor for fUTI. Therefore, our study was conducted to clarify the risk for fUTI in children without an anatomically based risk factor, such as CAKUTs including VUR. This information was added in the introduction section in the revised manuscript.

  1. The authors conclude that “Dysbiosis of gut microbiota during infancy can increase the risk of developing fUTI.” As this is a cross-sectional study, cause-and-effect relationship is hard to be proven. Although stool was collected before antibiotics administration, the changes in gut microbiota composition and UTI exist concurrently. In other words, fUTI might be a risk for gut microbiota dysbiosis. Please be careful to not overstate your point or make absolute conclusion that are not supported by evidence.

Response

We agree with you that this study design cannot prove a causal relationship between fUTI and dysbiosis in gut microbiota. Therefore, this point has been added to the limitations and discussed with related citations. In addition, with regard to the title and conclusion, we have replaced them with weaker expressions.

Minor issues:

  1. Table 1. Please add weeks to gestational age.

Response

We have added “weeks” to gestational age in Table 1. Thank you.

  1. Figure 1. Usually, the p value should be expressed to 3 digits. Please change p value to <0.001.
  2. Figure 2. Usually, the p value should be expressed to 3 digits. Please change p value to <0.001.

Response

We have changed p values expressed in 3 digits to p<0.001 throughout the revised manuscript. Thank you.

Reviewer 2 Report

In the article entitled "Dysbiosis is a risk factor for the development of febrile urinary tract infection in infancy", the authors, analyzing the intestinal microbiota of infants, try to answer the question whether disturbances in its composition may be the cause of a greater susceptibility to urinary tract infections. The work was well planned and described. The obtained results indicate differences in the microbiota in infants suffering from infections consisting of lower diversity and a significantly higher percentage of the Escherichia-Shigella genus. This work touches on the current trend of research on the analysis of the impact of intestinal microbiota disorders. Below are my notes and comments to the authors.

- the authors report that in most cases, infections are caused by E. coli, therefore finding a higher percentage of Escherichia-Shigella species in their microbiota is predictable. Does the result in which the infectious agent was Citrobacter differ from others?

- in the case of such small patients, can we talk about dysbiosis or rather about abnormal development of microbiota?

- in the discussion, it would be worth referring not only to the differences and importance of individual species, but also to the degree of microbiota diversity

- please unify the name of the microorganism in the text - Clostridioides difficile

Author Response

Reviewer 2

Comments and Suggestions for Authors

In the article entitled "Dysbiosis is a risk factor for the development of febrile urinary tract infection in infancy", the authors, analyzing the intestinal microbiota of infants, try to answer the question whether disturbances in its composition may be the cause of a greater susceptibility to urinary tract infections. The work was well planned and described. The obtained results indicate differences in the microbiota in infants suffering from infections consisting of lower diversity and a significantly higher percentage of the Escherichia-Shigella genus. This work touches on the current trend of research on the analysis of the impact of intestinal microbiota disorders. Below are my notes and comments to the authors.

Response

We thank the reviewer for this careful assessment of our manuscript and the useful comments provided. The comments of the reviewers were highly insightful and enabled us to greatly improve the quality of our manuscript. We have carefully revised our paper in response to these suggestions. In the following pages are our point-by-point responses to each of the reviewers’ comments. Modified sections are indicated by bold font and underlining in the revised manuscript.

  1. The authors report that in most cases, infections are caused by E. coli, therefore finding a higher percentage of Escherichia-Shigella species in their microbiota is predictable. Does the result in which the infectious agent was Citrobacter differ from others?

Response

Thank you for your important comment. It has been reported that Citrobacter is more likely to cause urinary tract infections in children with CAKUTs or neurological impairment. However, we excluded the patients with CAKUTs and neurological impairment in the current study. There were no obvious differences in clinical findings between this case and the other 27 cases. At present, the reason why one patient was infected with Citrobacter remains unknown.

  1. In the case of such small patients, can we talk about dysbiosis or rather about abnormal development of microbiota?

Response

Thank you for your important remark. We agree with your comment that the expression “abnormal development of gut microbiota” is appropriate for the infants in whom the gut microbiota has not yet been established. In accordance with your comment, we have replaced the expression "dysbiosis" with "abnormal development of gut microbiota" in the appropriate places.

  1. In the discussion, it would be worth referring not only to the differences and importance of individual species, but also to the degree of microbiota diversity

Response

We agree with your comment that the degree of microbiota diversity is also important when assessing dysbiosis. Therefore, we have added another analysis for alpha diversity, namely, Chao index, in addition to Shannon index, in the revised manuscript (Results section and Figure 1).

Furthermore, the significance of low-diversity microbiota has been stressed, as well as the differences of individual species, by adding the following sentences in the discussion section in the revised manuscript: “For alpha diversity, both Shannon and Chao indices were significantly lower in the fUTI group than in the HC group. The Shannon index represents the evenness, while the Chao index indicates the richness of microbiota. In low-diversity microbiota, relative abundance of beneficial bacteria which ferment complex sugars to short chain fatty acids including butyrate markedly decrease [Kriss M, et al. Low diversity gut microbiota dysbiosis: drivers, functional implications and recovery. Curr Opin Microbiol. 2018 Aug;44:34-40. doi: 10.1016/j.mib.2018.07.003. Epub 2018 Jul 20. PMID: 30036705]. Butyrate has been shown to have both local and systemic anti-inflammatory effects and thus its loss may mediate immune phenotypes in disease. Thus, it might be considered that fUTI is partly attributed to the low-diversity microbiota.”

Thank you very much for your important comment.

  1. Please unify the name of the microorganism in the text - Clostridioides difficile

Response

Thank you for your important remark.

We have changed “Clostridium difficile” to “Clostridioides difficile” on line 338.

Round 2

Reviewer 1 Report

Authors answered questions and comments that I have addressed to them by revision of the previous version of manuscript and included corresponding changes (based also on my comments) to the revised manuscript.